# Application of Nicotinamide to Culture Medium Improves the Efficiency of Genome Editing in Hexaploid Wheat

**DOI:** 10.3390/ijms24054416

**Published:** 2023-02-23

**Authors:** Wanxin Wang, Peipei Huang, Wenshuang Dai, Huali Tang, Yuliang Qiu, Yanan Chang, Zhiyang Han, Xi Li, Lipu Du, Xingguo Ye, Cheng Zou, Ke Wang

**Affiliations:** 1Institute of Crop Science, Chinese Academy of Agricultural Sciences, Beijing 100081, China; 2National Key Facility of Crop Resources and Genetic Improvement, Chinese Academy of Agricultural Sciences, Beijing 100081, China

**Keywords:** nicotinamide, wheat, CRISPR/Cas9, histone deacetylase inhibitors, editing efficiency, base editing

## Abstract

Histone acetylation is the earliest and most well-characterized of post-translation modifications. It is mediated by histone acetyltransferases (HAT) and histone deacetylases (HDAC). Histone acetylation could change the chromatin structure and status and further regulate gene transcription. In this study, nicotinamide, a histone deacetylase inhibitor (HDACi), was used to enhance the efficiency of gene editing in wheat. Transgenic immature and mature wheat embryos harboring a non-mutated *GUS* gene, the *Cas9* and a *GUS*-targeting sgRNA were treated with nicotinamide in two concentrations (2.5 and 5 mM) for 2, 7, and 14 days in comparison with a no-treatment control. The nicotinamide treatment resulted in *GUS* mutations in up to 36% of regenerated plants, whereas no mutants were obtained from the non-treated embryos. The highest efficiency was achieved when treated with 2.5 mM nicotinamide for 14 days. To further validate the impact of nicotinamide treatment on the effectiveness of genome editing, the endogenous *TaWaxy* gene, which is responsible for amylose synthesis, was tested. Utilizing the aforementioned nicotinamide concentration to treat embryos containing the molecular components for editing the *TaWaxy* gene, the editing efficiency could be increased to 30.3% and 13.3%, respectively, for immature and mature embryos in comparison to the 0% efficiency observed in the control group. In addition, nicotinamide treatment during transformation progress could also improve the efficiency of genome editing approximately threefold in a base editing experiment. Nicotinamide, as a novel approach, may be employed to improve the editing efficacy of low-efficiency genome editing tools such as base editing and prime editing (PE) systems in wheat.

## 1. Introduction

The development and application of genome editing technology has provided important technical support for plant functional gene research and crop genetic improvement. Compared with other developed genome editing technologies, CRISPR/Cas systems have the advantages of simple operation and high efficiency. Since the first application of CRISPR/Cas9 genome editing technology in eukaryotes in 2013 [1,2], it has become a revolutionary tool in the life sciences. The CRISPR/Cas9 system has been widely used for genome engineering in diverse eukaryotic species. Now, it is becoming a powerful biology tool for human gene therapy and plant breeding. CRISPR/Cas9 is mainly applied in crop genome editing for studies of gene function and genetic improvement.

Editing efficiency is an important issue for the CRISPR/Cas9 technique. There are several factors which normally influence efficiency. First, different gene editing tools and Cas9 variants have different editing efficiency levels. The editing efficiencies of SpCas9, CjCas9, SaCas9, StCas9, NmCas9, Cpfl, SpCas9NG, XCas9, SPG, and SPR were different [3,4,5,6] and up until now, the CRISPR/Cas9 system was the most efficient and widely used solution for genome editing. Second, editing efficiency is also influenced by the the integration locations and expression levels of the *Cas9* and targeted sgRNA in the genome of transgenic plants ae well as the structures of the sequences of the sgRNA in the genome [4]. Third, the editing efficiency of the CRISPR systems differs in different species. For instance, the editing efficiency of Cpf1 reached 75% in rice [7] but only 3.1% in wheat [8]. It is generally believed that genome editing is more difficult in polyploid than diploid plants because there is only one copy of a functional gene in diploid plants and multiple copies in polyploid ones. Target traits are only displayed when their multiple controlled genes are edited simultaneously in a single polyploid plant. For example, although the editing efficiencies of wheat *TaMTL* and *TaWaxy* genes were up to 57.5% and 80.5%, respectively, the simultaneous editing efficiencies at their three loci were only 12.9% and 32.2%, respectively [8]. Finally, environmental conditions and artificial treatments could affect the editing efficiency of CRISPR. It was reported that high-temperature treatment increased the editing efficiency of Cpf1 in *Arabidopsis*, rice, maize, and wheat [9,10].

Previous reports reported that chromatin structure could influence the genome editing efficiency in different eukaryotic cells [11,12]. In general, eukaryotic genomic DNA is wrapped around histones, which first form nucleosome structures and further compact into higher-order chromatin structures [13,14]. Chromatin structures actually involve an alternation between open and closed states. The regions which have low concentrations of histones, or naked DNA, are called open chromatin; DNA replication and gene transcription occur in these regions [15]. Thereby, the activity of the Cas9 protein is significantly different between the closed area winding the nucleosome and the open area, and Cas9 cannot bind to the DNA on the nucleosome [14,16,17]. The binding site of Cas9 in mammals was found that the binding of Cas9 to target genes mainly occurred in the open chromatin region [11]. In fact, the efficiency of CRISPR/Cas9-induced insertion and deletions (indels) in human cells was higher at the sites in open chromatin regions than in closed chromatin regions [18]. Based on the published genomic data and other available information on rice, the editing efficiency of CRISPR/Cas9 in open regions was higher than that in closed regions [19].

To our knowledge, chromatin remodeling can make the core histone octamer slide on the DNA strand and change the distance between nucleosomes or the conformation of nucleosomes by replacing histone variants or removing the nucleosomes [20]. Thus, gene replication, recombination, transcription, and expression can be regulated through exposure of the promoter and enhancer or DNA replication of target genes [20]. There are two main types of enzymes that regulate the process of chromatin remodeling: ATP-dependent chromatin remodelers and histone modifiers. It has been demonstrated that the interaction between histone acetylation modifications and transcription factors is beneficial, promoting chromatin opening, and that it could potentially improve the editing efficiency of the CRISPR system [21]. Histone acetylation, as the earliest and most well-characterized of post-translation modifications, plays a major role in chromatin remodeling. Histone acetylation, which is mediated by histone acetyltransferases (HAT) and histone deacetylases (HDAC), can change chromatin structures and regulate gene transcription, DNA replication, and repair [22].

Histone deacetylase inhibitors (HDACis) were used to treat the HEK293T cell line by adding acetyl groups into histones to promote the opening of chromatin, and the editing efficiency of CRISPR/Cas9 system was improved [23]. Sodium butyrate and nicotinamide are two commonly applied histone deacetylase inhibitors (HDACis). Sodium butyrate is a short-chain fatty acid that inhibits the *Arabidopsis* RPD3/HDA1 in *HDAC* gene family [24]. Nicotinamide, as a derivative of vitamin B3 and a product of the SIR2 HDAC family deacetylation reaction, inhibits the function of HDACs in mammals and yeast [25,26]. The exposure of maize cells to HDACis led to open chromatin structures in the region of target genes and further improved the efficiency of oligonucleotide-directed mutagenesis (ODM) [27]. In our previous study, wheat seedlings were treated with nicotinamide, and transcriptome analyses found that nicotinamide treatment would change the expression of chromatin state-related genes [28]. In this study, we investigated the effects of nicotinamide in different concentrations on several wheat transgenic tissues, including immature embryos, mature embryos, and calluses, to enhance genome editing efficiency through alterations in chromatin status. Our results provide a new strategy to improve wheat genome editing efficiency.

## 2. Results

### 2.1. Effect of Nicotinamide Treatment of Transgenic Wheat Immature Embryos on the Mutation Efficiency of GUS Gene

The T_1_ transgenic plants harboring the *Cas9* and a *GUS*-targeting sgRNA generated earlier by the pMWB110-SpCas9-TaU3-GUS vector, denoted GUC plants in which the *GUS* gene was not edited [8], were planted in a greenhouse for this experiment. In order to confirm if the T_1_ plants still carry the *bar* gene linked to *Cas9*, PCR was performed (Appendix A), which resulted in detection of the *bar* gene in 48 plants out of the 62 plants tested. Then, the lack of the mutations in the *GUS* gene in the *bar*-positive plants was tested with PCR-RE (Appendix A). No mutation in the *GUS* gene was detected in any of the transgenic plants tested.

In total, 240 immature embryos, collected from the positive T_1_ plants, were cultured with 2.5 and 5.0 mM nicotinamide for 2, 7, and 14 days designated as 2-2.5-G, 2-5-G, 7-2.5-G, 7 -5-G, 14-2.5-G, and 14-5-G, respectively, and without nicotinamide designated as 2-G, 7-G, and 14-G. When the cultured T_2_ plants were transplanted into pots, they were all analyzed for the presence of the *bar* gene, and 173 plants were confirmed to contain the *bar* gene as well as *Cas9/sgRNA* expression cassette. Then, the mutations of *GUS* gene in the positive plants were tested by PCR-RE (Appendix A). There were no edited plants in 2-G, 2-2.5-G, and 2-5-G groups (Appendix A), while 3 and 4 mutations were detected in groups 7-2.5-G and 7-5-G with mutation efficiencies of 16.7% and 21.1%, respectively (Appendix A). Furthermore, 9 and 4 mutations were found in groups 14-2.5-G and 14-5-G with editing efficiencies of 36.0% and 17.4%, respectively (Table 1). Moreover, one homozygous mutant plant was detected in group 14-2.5-G. The mutation efficiency of *GUS* after 2.5 mM nicotinamide treatment for 14 d was the highest (36.0%).

To further confirm the new mutants after nicotinamide treatment, the *GUS* gene in the plants was sequenced and analyzed. The GUS staining showed that the homozygous mutant did not exhibit blue coloring, while the other heterozygous mutants still expressed the *GUS* gene (Figure 1a). The types of mutations in *GUS* gene in the new mutants were mainly 1 bp, 3 bp, 5bp, and 12 bp deletions after sequencing (Figure 1b).

### 2.2. Effect of Nicotinamide Treatment to Transgenic Wheat Immature Embryos on the Mutation Efficiency of TaWaxy Genes

In order to further confirm the effects of nicotinamide treatment, the transgenic T_0_ plants, which had no mutations within endogenous *TaWaxy* genes and harbored *Cas9* and a *TaWaxy*-targeting sgRNA cassette (denoted WUC plants), were analyzed for the presence of *bar* genes by PCR and mutations of *TaWaxy* genes by PCR-RE (Appendix A). The results indicated that there was still no mutation in the *TaWaxy* gene in the plants testing positive. Then, the immature embryos of T_0_ WUC plants were cultured on 2.5 and 5.0 mM nicotinamide and control medium for 14 days. There were 58, 71, and 64 embryos for 14-W1, 14-2.5-W1, and 14-5-W1 groups, respectively. After the cultured plants were transplanted into pots, 44, 53, and 48 positive plants were confirmed in 14-W1, 14-2.5-W1, 14-5-W1, respectively, after PCR testing for the *bar* gene. Three pairs of specific primers were used to detect mutations of *TaWaxy* genes on chromosomes 4A, 7A, and 7D by PCR-RE (Appendix A). There was still no mutation detected in 14-W1, while 7 and 5 mutations were detected in 14-2.5-W1 and 14-5-W1, respectively, with mutation efficiencies of 13.2% and 10.4% (Table 2). In group 14-2.5-W1, 2, 3, and 5 mutations were detected at the loci on 4A, 7A, and 7D, respectively, and the simultaneously mutation frequencies for one, two, and three loci were 7.5%, 5.7%, and 0%, respectively (Table 2). In group 14-5-W1, 1, 2, and 4 mutations were detected at the three loci with editing efficiencies of 2.1%, 4.1%, and 8.4%, respectively, and the simultaneous mutation frequencies for one, two, and three loci were 6.3%, 4.2%, and 0%, respectively (Table 2). Additionally, it was found that the editing efficiency in 14-2.5-W1 was slightly higher than that in 14-5-W1.

Considering that nicotinamide treatment can significantly improve editing efficiency in transgenic and unedited wheat plants and obtain the edited plants at up to two loci simultaneously, we treated the edited plants at one locus to obtain edited plants at three loci. Hence, a total of 42 immature embryos of the transgenic and edited wheat plants which had either one mutation among the three loci on chromosomes 4A, 7A, and 7D were treated on 2.5 and 5.0 mM nicotinamide and control for 14 d. There were only 12, 13, and 8 positive cultured plants, respectively, confirmed in groups 14-W2, 14-2.5-W2, and 14-5-W2 after *bar* gene detection. Based on the detection results by PCR-RE (Appendix A), new mutation efficiencies of 8.3%, 30.8%, and 25.0% were achieved in 14-W2, 14-2.5-W2, and 14-5-W2, respectively (Table 3). In 14-W2, only one plant had two mutations, at the loci on 4A and 7A, in which the mutation on 4A was newly edited. In 14-2.5-W2, three plants with two locus mutations and one plant with three loci mutations were identified, respectively, with a new mutation efficiency of 30.8%. In 14-5-W2, two loci mutations were found in two plants. It could be summarized that the 2.5 mM nicotinamide was the best concentration to treat immature wheat embryos with the goal of creating new editing loci.

To further confirm the new mutants after nicotinamide treatment on the immature embryos, the *TaWaxy* gene in the candidate mutant plants was sequenced and analyzed. The types of mutations in *TaWaxy* gene mutants were mainly 1 bp, 4 bp, 7 bp, 13 bp, 533 bp deletion mutations and a 1 bp insertion mutation. (Figure 2).

### 2.3. Effects of Nicotinamide Treatment of Transgenic Mature Wheat Embryos on the Mutation Efficiency of TaWaxy Genes

In order to explore the effects of nicotinamide treatment of transgenic mature wheat embryos on the mutation efficiency of *TaWaxy* genes, we harvested the seeds of T_1_ transgenic plants with expression editing cassettes and without mutations. Their mature embryos were cultured on control medium and treatment medium containing 2.5 and 5.0 mM nicotinamide for 14 d, and then the in-vitro-treated plantlets were moved into pots after sufficient growth. After *bar* gene detection, 21, 15, and 8 positive plants were identified in 14-W3, 14-2.5-W3, and 14-5-W3, respectively. Furthermore, based on PCR-RE detection (Appendix A), no mutations were detected in 14-W3, but two edited plants were found in 14-2.5-W3 and one edited plant was found in 14-5-W3, with mutation efficiencies of 13.3% and 12.5%, respectively (Table 4). When the mutations were detected by sequencing, the results showed that the mutation types in *TaWaxy* gene in the mutants were mainly 1 bp and 4 bp deletions and a 1 bp insertion (Figure 3).

### 2.4. Determination of Available Concentration of Nicotinamide Treatment during Wheat Genetic Transformation Steps

To confirm if nicotinamide treatment was able to influence wheat genetic transformation, different dosages of nicotinamide (0, 2.5, and 5 mM) were added into the WLS-Res medium in the genetic transformation of NGT2 vector using the immature embryos of Fielder by *Agrobacterium*. The results clearly showed that the addition of nicotinamide to medium inhibited the growth of callus (Figure 4). With increased nicotinamide concentration, callus growth was significantly negatively influenced, and callus differentiation capacity was also greatly decreased. Obviously, the concentration of nicotinamide was negatively correlated with the efficiency of wheat transformation (Table 5). The efficiency reached up to 93.0% for NGT2 without nicotinamide, while the efficiency levels were only 54.6% and 18.4% for NGT2-2.5 and NGT2-5, respectively. The application of 5 mM nicotinamide led to an extremely low transformation efficiency; therefore, this concentration should not be applied during wheat genetic transformation.

### 2.5. Application of Nicotinamide in Wheat Transformation for Improving Mutation Efficiency by Base Editing

An expression vector, ABENG-Wx, was used for base editing in wheat genetic transformation to investigate the effects of nicotinamide application, with available concentrations, on the mutation efficiency of base editing. Since there were only two copies of *TaWaxy* genes in Fielder on chromosomes 7A and 7D, another copy of *TaWaxy-7B* did not need to be detected in the transgenic plants. As shown in Table 6, only one mutant was found in 30 transgenic plants in the control experiment (without nicotinamide treatment) and the efficiency of the target gene (by base editing) was only 3.3%. A total of six mutants were detected in the treatment experiment with 2.5 mM nicotinamide, and the editing efficiency was increased nearly threefold (9.7%). Moreover, *TaWaxy-7A* and *TaWaxy-7D* were found to be simultaneously edited in two mutants. The results of sequencing shown that all of mutations occurred at the 4th or 7th base of the sgRNA, only one transgenic plants have the mutation at both positions simultaneously (Figure 5).

## 3. Discussion

### 3.1. The Factors Affect the HDACi Treatment

In fact, many factors influence the effect of HDACi treatment, listed as follows: (1) The types of HDACi. Sodium butyrate and nicotinamide were two kinds of HDACi. Bond et al. found that the expression of the *VIN3* gene in *Arabidopsis* was increased when the plants were exposed to sodium butyrate and nicotinamide, indicating that HDACi could relax chromatin [29]. It was also found that the pretreatment of maize cells with either sodium butyrate and nicotinamide had a positive effect on the efficiency of oligonucleotide-directed mutagenesis [27]. In the two chemicals, treatment with nicotinamide had a less severe effect on plant growth than sodium butyrate in *Arabidopsis* [29]. Additionally, sodium butyrate has been proven to exert a negative effect on wheat tissues [28]. Therefore, it is better to treat wheat tissues using nicotinamide. (2) The concentration of nicotinamide. It was found that the effect of nicotinamide treatment at 5 mM is 2.5 fold higher than that at 2.5 mM in *Arabidopsis* [29]. Moreover, when nicotinamide (1–5 mM) was used in maize cells, the editing efficiency enhanced with the increased concentration [27], while our results showed that the effect of 2.5 mM nicotinamide treatment on editing efficiency is optimal (Table 1 and Table 2). (3) The time of nicotinamide treatment. The times normally influence the effect of nicotinamide treatment (Table 1 and Table 5). In this study, we obtained the best editing results using nicotinamide to treat wheat tissues for 7 d (Table 1). (4) The tissues used for nicotinamide treatment. It was reported that the nicotinamide treatment showed a good effect on seedlings and cells [27,29]. In this study, the editing efficiency was proved to be improved by treating wheat callus, mature, and immature embryo with nicotinamide. (5) Species. Different plant species may also lead to different effects of nicotinamide treatment. (6) Other factors. In this study, the treatment effect of nicotinamide for the *GUS* gene is higher than that for the *TaWaxy* gene. This might be caused by different sgRNA or even different insertion sites of transgenes which might lead to different expression of the *Cas9* gene.

### 3.2. Application Prospects of Nicotinamide in Wheat for Efficient Genome Editing

Although the editing efficiency of CRISPR is high in some diploid plants, it is still somewhat difficult to obtain mutants in which all the target allelic genes are simultaneously edited in polyploid plants. For example, in wheat, the editing efficiency reached 51.7% for the three *TaNud* alleles as a whole but only 8.5% for *TaNud-7D* alone [30]. When we edit some genes in wheat, there is no mutation, and generally we need to do transformation experiments again. In this study, we found that nicotinamide treatment can increase the editing efficiency to 36% from 0% (Table 1) in the next generation of non-edited transgenic plants. Therefore, nicotinamide treatment can be used to treat the non-mutation transgenic plants for obtaining the mutants, which saves time and cost. Normally, if there is only single chromosome mutation happening in different plants, crosses need to be made with each other for obtaining triple mutations. In this study, the efficiency of triple mutations was increased to 7.7% from 0% after nicotinamide treatment on single-chromosome mutations under optimal conditions (Table 3). Therefore, the application of nicotinamide treatment is an available choice for improving editing efficiency in wheat.

Adenine base editors (ABEs) and cytosine base editors (CBEs) are two major developments in base editing tools. Recently, several evolved ABE8e systems have been reported to increase A-to-G editing efficiency in plants [31,32,33,34]. ABE8e were fused with SpCas9 nickase variants (SpCas9-NG) that recognized the PAM NGN to form ABE8e-NG and led to an average editing efficiency of 59.3% for the tested NGN target sites in rice [35]. In the present study, the editing efficiency of ABE8e-NG was only 3.33% in wheat, but it could be increased nearly threefold after nicotinamide treatment. Therefore, the effect of nicotinamide treatment on the editing efficiency by base editing is very positive. The prime editing (PE) system is another recent genome editing tool and the latest research hotspot. However, as of now, its efficiency is still very low [36,37]. The application of nicotinamide treatment might provide an effective strategy to improve the editing efficiency by PE system. Therefore, we assumed that nicotinamide treatment would have a great application potential for the editing of inefficient target sites in wheat using base editing and PE systems.

## 4. Materials and Methods

### 4.1. Plant Materials

The plant materials used in this study included T_1_ homozygous transgenic wheat line H29 in the genetic background of CB037 which carried an active *GUS* gene (called first time transformation), T_1_ transgenic plants of non-mutated *GUS* containing *Cas9* and the sgRNA for *GUS* in the genetic background of H29 (called second transformation), and T_0_ transgenic plants transformed with pMWB110-SpCas9-TaU3-Waxy vector in the genetic background of Ningchun4 for nicotinamide treatment using their immature embryos, in which the target genes were not edited or partially edited. The aforementioned materials were obtained from our previous study [8]. Additionally, a spring wheat cultivar, Fielder, maintained in our group was also used to study the influence of nicotinamide treatment on callus growth and transformation efficiency. All the plants were grown in pots in a growth chamber maintained at 24 °C under 16/8 h light/dark conditions with 300 μmol m^−2^ s^−1^ light intensity and 45% humidity.

The immature embryos of the T_0_ transgenic plants were treated with nicotinamide, and their mature embryos from the harvested seeds of no-edited plants were treated with nicotinamide again. Then, the effects of nicotinamide treatment on subsequent editing efficiency were investigated.

### 4.2. Design of Wheat Mature and Immature Embryos with Nicotinamide Treatment

About 14 days post-anthesis (DPA), the immature wheat grains were sampled, sterilized with 70% ethanol for 1 min and 15% sodium hypochlorite for 10 min and then washed three times with sterile water. The immature wheat embryos were isolated and cultured on 1/2 MS medium (2.215 g L^−1^ MS powder, 20 g L^−1^ sucrose, 0.5 g L^−1^ MES, and 3 g L^−1^ phytagel, pH 5.8) with two concentrations of nicotinamide (2.5 and 5.0 mM, N0636, Sigma, Louis, MO, USA) at 2, 7, and 14 d in order as well as a non-treated control in which the germinated embryos in the treatments with nicotinamide for 2 and 7 d were promptly moved onto the control medium for a total culture period of 14 d. Then, the plantlets were transplanted into pots with soil for further growth.

The mature wheat grains were surface sterilized with 70% ethanol for 1 min and with 15% sodium hypochlorite for 20 min, washed with sterile water three times, and then immersed in sterile water before being placed in a shaker overnight. The slightly germinated seeds were sterilized with 15% sodium hypochlorite for 20 min and washed with sterilized water three times. The mature embryos were separated from the seeds and cultured on 1/2 MS medium with nicotinamide at the same levels and time intervals above.

### 4.3. Detection of Mutations

Genomic DNA was extracted from the leaves of candidate wheat mutant plants using a Nuclean Plant Genomic DNA Kit (CW0531M, CWBIO, Jiangsu, China). The specific primers were used to amplify the targeted genes of *bar*, *GUS,* and *TaWaxy*, in order (Appendix A). In order to confirm that the candidate plants contained the genome editing cassette, the *bar* gene was detected first. Then, the PCR-restriction enzyme (PCR-RE) method was applied to detect the mutants, in which the specific PCR products were digested at 37 °C for 2 h in the 20 μL reaction buffer consisted of the corresponding restriction enzymes (1 U each) and 10 μL PCR products. The restriction enzymes of *SnaBI* and *BglII* were used to detect the *GUS* and *TaWaxy* gene mutants, respectively. The resultant products were separated in a 1% agarose gel and visualized using a GelDoc XR System (BioRad, Hercules, CA, USA). Three types of band patterns were found in the PCR-RE assay; heterozygous monoallelic mutants gave three bands, biallelic mutants gave only the largest band, and nonmutants and wild-type (WT) plants gave two completely digested bands. The PCR products were directly sequenced at Sangon Biotech (Shanghai, China) for analysis of the mutations induced by base editing.

### 4.4. Base Editing Vector Construction

The pWMB110-SpCas9 containing the bar gene as a selection marker for generating transgenic plants and the maize (Zea mays) ubi promoter for driving the expression of *Cas9* gene had previously been constructed by our laboratory [8]. The sequences of ABE8e and SpCas9-NG [35] were optimized based on the wheat genome sequencing data and then inserted the pMWB110-SpCas9 vector to replace the SpCas9. In order to increase the transformation efficiency, a regeneration-related gene, TaWOX5, was introduced into the base editing vector to generate the pMWB110-ABE-NG-TaWOX5 plasmid [38]. The sgRNA1 (AAGACCAAGGAGAAGATCTA) and sgRNA2 (CTGGATGAAGGCCGGGATCCTGC) were designed on https://crispr.bioinfo.nrc.ca/WheatCrispr/, (accessed on 21 February 2023) based on the sequences of *TaWaxy* genes for base editing, and then the sgRNA controlled by TaU3 promoter was constructed onto plasmid pMWB110-ABE-NG-TaWOX5 to generate ABENG-Wx and NGT2 vectors according to the method described by Liu et al. [8], respectively. The base editing vector was transformed into *Agrobacterium* strain C58C1 for wheat transformation.

### 4.5. Wheat Transformation

Immature wheat embryos were isolated and underwent *Agrobacterium*-mediated transformation to obtain transgenic plants following the protocol described by Wang et al. [38]. In brief, immature wheat embryos were incubated with *Agrobacterium* strain C58C1-harboring a transformation vector for 5 min in WLS-inf medium (1/10 Linsmaier and Skoog (LS) salts, 1/10 Murashige and Skoog (MS) vitamins, glucose 10 g L^−1^, 2-(N-morpholino) ethanesulfonic acid (MES) 0.5 g L^−1^, and acetosyringone (AS) 100 μM, pH 5.8) at room temperature and co-cultivated for 2 d on WLS-AS medium (WLS-inf medium plus AgNO_3_ 0.85 mg L^−1^, CuSO_4_·5H_2_O 1.25 mg L^−1^ and agarose 8 g L^−1^) with scutellum facing upwards at 25 °C in darkness. After co-cultivation, embryonic axes were removed with a scalpel, and the scutella were transferred onto plates containing WLS-Res medium (LS salts, MS vitamins, 2,4-d 0.5 mg L^−1^, picloram 2.2 mg L^−1^, AgNO_3_ 0.85 mg L^−1^, ascorbic acid 100 mg L^−1^, carbenicillin 250 mg L^−1^, cefotaxime 100 mg L^−1^, MES 1.95 g L^−1^, and agarose 5 g L^−1^) for delay culture for 7 d with different nicotinamide concentrations. Afterward, the tissues were moved onto WLS-P5 medium (WLS-Res medium plus phosphinothricin (PPT, Sigma, no. 45520) 5 mg L^−1^ without cefotaxime) for callus induction. Two weeks later, the calli were shifted onto WLS-P10 medium (WLS-Res medium plus PPT 10 mg L^−1^ without cefotaxime) for 3 weeks in darkness. The embryogenic calli were then differentiated on LSZ-P5 medium (1/2MS medium containing PPT 5 mg L^−1^) at 25 °C with 100 μmol/m^2^/s light. The regenerated shoots were transferred into cups filled with LSF-P5 medium (LSZ-P5 medium plus IBA 1 mg L^−1^) for elongation and root formation. The plantlets with well-developed root systems were transplanted into pots and cultivated in a growth chamber.

## Figures and Tables

**Figure 1 ijms-24-04416-f001:**
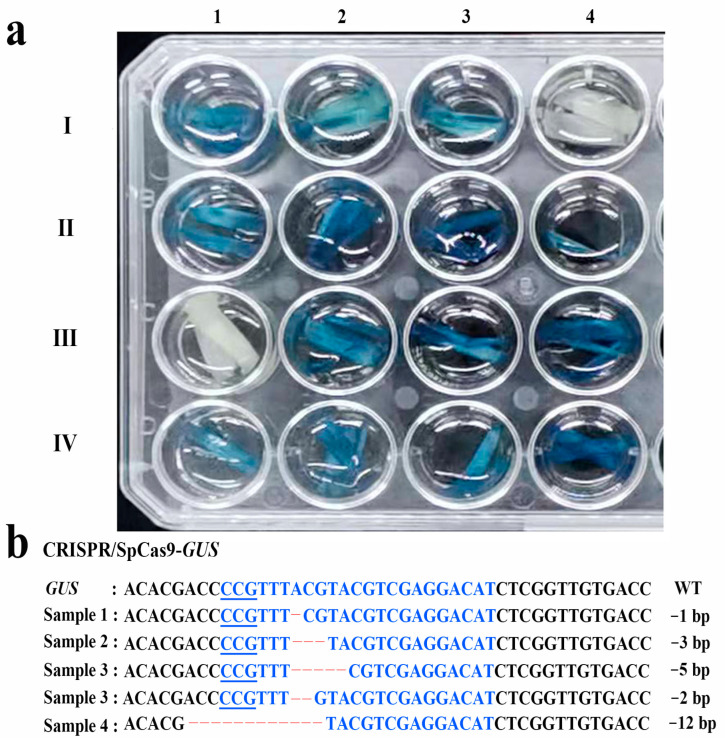
GUS staining and sequence analysis in the new mutant wheat plants. (**a**) The expression of GUS protein in mutants. I1-I3: the mutants from groups 7-2.5-G. I4: non-transgenic variety CB037. II1-II4: the mutants from the groups 7-5-G. III1-III4: the mutants from the groups 14-2.5-G. IV1-IV3: the mutants from the groups 14-5-G. IV4: the transgenic line H29 with *GUS* gene. III1: plant with biallelic mutations (the other mutants are heterozygous mutations). (**b**) Deletion mutations in the *GUS* gene in the mutants. The PAM region is underlined; blue letters indicate sgRNA sequences, and dashed lines represent nucleotide deletions.

**Figure 2 ijms-24-04416-f002:**
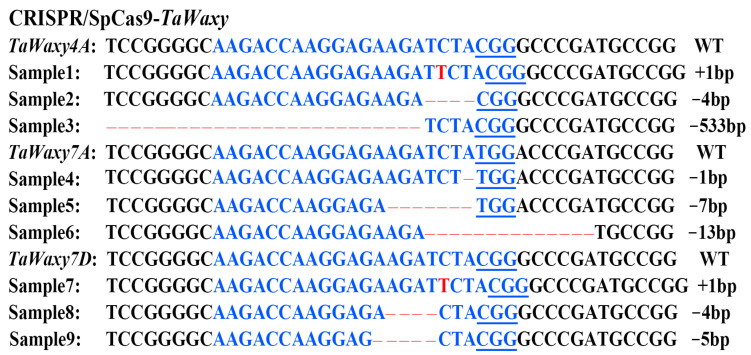
InDel mutations of the *TaWaxy* gene in the mutants generated from the immature embryos under nicotinamide treatment in vitro. The PAM region is underlined; blue letters indicate sgRNA sequences, red letters indicate insertions, and dashed lines represent nucleotide deletions.

**Figure 3 ijms-24-04416-f003:**
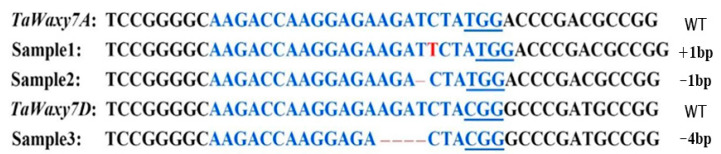
InDel mutations of the *TaWaxy* gene in the candidate mutants generated from the mature embryos after nicotinamide treatment. The PAM region is underlined; blue letters indicate sgRNA sequences; red letters indicate insertions; dashed lines represent nucleotide deletions.

**Figure 4 ijms-24-04416-f004:**
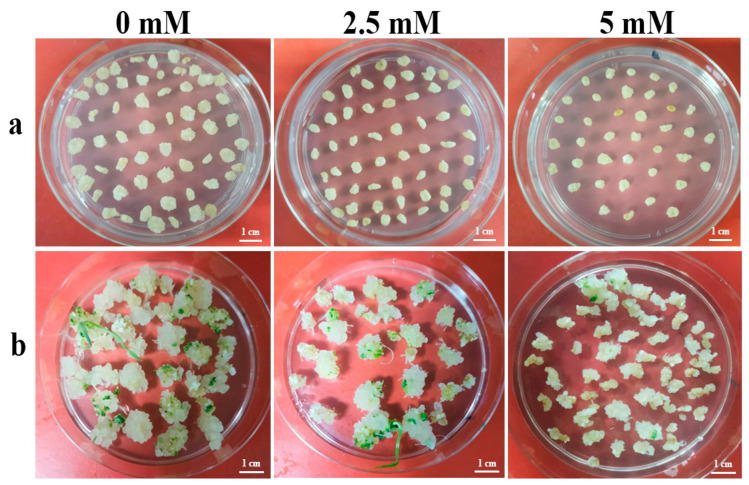
Callus growths and shoot regeneration status on the selection culture medium containing different concentrations of nicotinamide during wheat genetic transformation. (**a**) Callus growth status on WLS-P5 medium. (**b**) Shoot regeneration status on LSF-P5 medium.

**Figure 5 ijms-24-04416-f005:**
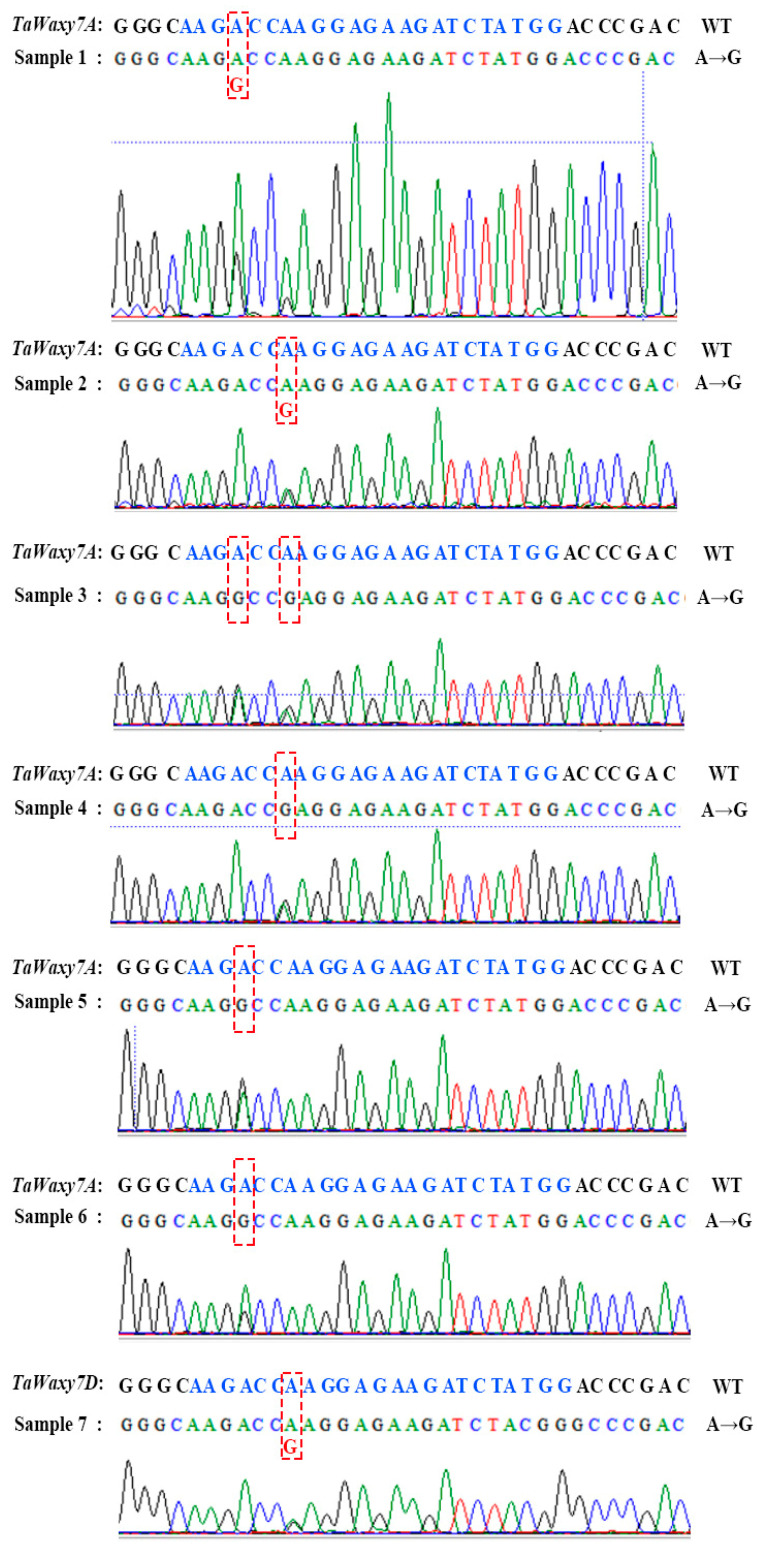
The sequencing results of base editing for the *TaWaxy* gene mutants. Sample 1: the mutants from the control group. Samples 2–7: the mutants from the nicotinamide treatment group.

**Table 1 ijms-24-04416-t001:** The editing efficiency of *GUS* gene after nicotinamide treatment of immature embryos of the double transgenic parent plants which carry an active *GUS* gene (homozygous, first transformation) and genome editing vectors (hemizygous, second transformation) in different concentrations.

Groups	Nicotinamide Concentration (mM)	Treatment Time (d)	Plants Tested	*bar*-Positive	No. of Mutants	Editing Efficiency (%)
2-G	0	2	17	13	0	0
2-2.5-G	2.5	2	28	21	0	0
2-5-G	5	2	27	22	0	0
7-G	0	7	25	15	0	0
7-2.5-G	2.5	7	20	18	3 (He)	16.7
7-5-G	5	7	26	19	4 (He)	21.1
14-G	0	14	26	17	0	0
14-2.5-G	2.5	14	36	25	8 (He) + 1 (Bi)	36.0
14-5-G	5	14	35	23	4He	17.4

Note: Bi: biallelic mutation type; He: heterozygous mutation type.

**Table 2 ijms-24-04416-t002:** The editing efficiency of *TaWaxy* gene after nicotinamide treatment of the immature embryos of the transgenic plants which carry genome editing vectors (hemizygous) and without mutation in different concentrations.

Groups	Plants Tested	*Bar*-Positive Plants	Mutation Types at Different Locus on Chromosome			No. of Mutants	Editing Efficiency (%)	No. of the mutant Edited at One or Two Loci	
			4A	7A	7D			1	2
14-W1	58	44	0	0	0	0	0	0	0
14-2.5-W1	71	53	2He (3.7%)	3He (5.7%)	5He (9.4%)	7He	13.2	4 (7.5%)	3 (5.7%)
14-5-W1	64	48	1He (2.1%)	2He (4.1%)	4He (8.4%)	5He	10.4	3 (6.3%)	2 (4.2%)

Note: He: heterozygous mutation type.

**Table 3 ijms-24-04416-t003:** The editing efficiency of *TaWaxy* genes with new mutations after nicotinamide treatment on immature embryos of transgenic plants which carry genome editing vectors (hemizygous) and with single chromosome mutations.

Groups	Plants Tested	*Bar*-Positive Plants	Mutation Types at Different Locus on Chromosome			No. of New Mutants	Editing Efficiency (%)	No. of the Mutant Edited at Two or Three Loci	
			4A	7A	7D			2	3
14-W2	15	12	1He (8.3%)	0	0	1	8.3	1 (8.3%)	0
14-2.5-W2	17	13	1He (7.7%)	3He (23.1%)	1He (7.7%)	4	30.8	3 (23.1%)	1 (7.7%)
14-5-W2	10	8	1He (12.5%)	1He (12.5%)	0 0	2	25.0	2 (25%)	0

Note: He: heterozygous mutation type.

**Table 4 ijms-24-04416-t004:** The editing efficiency of *TaWaxy* gene after nicotinamide treatment of the mature embryos of the transgenic plants which carry genome editing vectors (hemizygous) and without mutation with different concentrations.

Groups	Plants Tested	*Bar* Positive Plants	Mutation Types at Different Locus on Chromosome			No. of Mutants	Editing Efficiency (%)	No. of the Mutant Edited at One or Two Loci	
			4A	7A	7D			1	2
14-W3	24	21	0	0	0	0	0	0	0
14-2.5-W3	18	15	0	1He (6.7%)	1He (6.7%)	2	13.3	2 (13.3%)	0
14-5-W3	9	8	0	1He (12.5%)	0	1	12.5	1 (12.5%)	0

**Table 5 ijms-24-04416-t005:** Effects of nicotinamide on the efficiency of *Agrobacterium*-mediated genetic transformation of wheat.

Vector	Nicotinamide Treatment (mM)	No. of Explants Transformed	No. of Positive Plants	Transformation Efficiency (%)
NGT2	0	57	53	92.98
NGT2	2.5	55	30	54.55
NGT2	5	38	7	18.40

Note: The genotype was Fielder and the nicotinamide was added in WLS-Res medium.

**Table 6 ijms-24-04416-t006:** The base editing efficiency of *TaWaxy* genes using nicotinamide treatment.

Vector	Nicotinamide Concentration (mM)	Target Loci	No. of Plants Identified	No. of Plants Edited	Mutation Efficiency (%)
ABENG-Wx	0	*TaWaxy-A*	30	1	3.33
		*TaWaxy-D*		0	0
		*TaWaxy-AD*		0	0
		Total		1	3.33
ABENG-Wx	2.5	*TaWaxy-A*	62	5	8.06
		*TaWaxy-D*		2	3.23
		*TaWaxy-AD*		2	3.23
		Total		6	9.68

## Data Availability

The data presented in this study are available in the article and Appendix A.

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
