# Peer review of "Application of Nicotinamide to Culture Medium Improves the Efficiency of Genome Editing in Hexaploid Wheat"

_ijms, 2023, doi:10.3390/ijms24054416_

Round 1

Reviewer 1 Report (Previous Reviewer 1)

This manuscript entitled “Application of Nicotinamide in Culture Medium Improves the Efficiency of Genome Editing in Wheat” had been revised at two times, so the quality was strongly improved. Some minor concerns should be revised or considered before publication.

1.     Throughout the manuscript, “sample” should be replaced with fitful term, which is specific regenerated plants.

2.     In the knockout of gene by Cas9 editing, all the control excluding one example showed no mutation, which misled reader that this gene editor is poorly potential to perform gene editing in wheat. Please check and explain.

3.     Line 132-133, here, authors appeared to compare all the treatments, not only the control group. Please revise it.

4.     The biallelic mutants should be needed to sequence for the sequence shown in Fig 2b.

Author Response

This manuscript entitled “Application of Nicotinamide in Culture Medium Improves the Efficiency of Genome Editing in Wheat” had been revised at two times, so the quality was strongly improved. Some minor concerns should be revised or considered before publication.

  1. Throughout the manuscript, “sample” should be replaced with fitful term, which is specific regenerated plants.

Response: Thanks for your suggestion. The “sample”  had been changed to “ transgenic plants” in the manuscript.

  1. In the knockout of gene by Cas9 editing, all the control excluding one example showed no mutation, which misled reader that this gene editor is poorly potential to perform gene editing in wheat. Please check and explain.

Response: These results are no problem. More than 50 genes were knockout by Cas9 per years in our lab,and many researches about gene editing in our lab have been published. In the detection for gene editing, we found that the different tillers maybe have different mutations (Zang et al. 2022). So if you only detect one leaf of one plant from T0 generation, you can found the new mutations in T1gerneration, but the new generation maybe not appeared in T1 but maybe had appeared in T0. We had finished many gene editing experiment, in our experience it is very difficult to generate new mutation in the next generation in wheat.

  1. Line 132-133, here, authors appeared to compare all the treatments, not only the control group. Please revise it.

Response: Thanks for your nice suggestion. The sentence about control group had been deleted.

  1. The biallelic mutants should be needed to sequence for the sequence shown in Fig 2b.

Response: Thanks for your suggestion. We have added the sequences of biallelic mutant in Fig 1b.

Reviewer 2 Report (Previous Reviewer 2)

Paper has been improved deeply. New figures have been included to address most of my issues. Therefore I recommend this work for publication.

Author Response

Paper has been improved deeply. New figures have been included to address most of my issues. Therefore I recommend this work for publication.

Response: Thanks for your positive comments.

Reviewer 3 Report (New Reviewer)

This manuscript reports a novel and interesting application of treating plants with nicotinamide to enhance genome editing efficiency. Poor editing efficiency is indeed a challenge in many crops, especially polyploids such as wheat. The authors have standardized various concentrations and treatment intervals of nicotinamide. They study the editing efficiency of nicotinamide in wheat callus, as well as mature, and immature embryos. They were able to edit GUS gene with single copy and the TaWaxy that has three copies on three different chromosome. It is appreciable indeed that the authors achieved enhanced editing efficiencies. However, there are some minor suggestions/queries -

Lines 77–79 - Please cite appropriate reference.

Lines 113-114 – The authors write about 48 and 64 plants, but the image has lesser lanes. Some lanes do not have bands, pl write the explanation in legend as ‘putatively transgenic’ and not ‘transgenic’.

Lines 114-116 – The result is not explained clearly. What is it that the authors want to convey? Please explain.

Lines 188-194 – The information is unclear. Is it that the authors want to convey that they repeat nicotinamide treatment on lines mutated at one locus in order to achieve mutations at two additional loci. If that is the case, nicotinamide works randomly and I wonder if it should be able to aid editing loci on multiple chromosomes during the first round of 14-day treatment itself.

Lines 311-312 – Cite the reference.

Lines 389-391 – Was the digestion performed after completing the PCR reaction or before?

Line 393 – Check 2%. I wonder if the authors used this concentration to separate bands of sizes 300-500 bp.

Lines 396-397 – Details of sequencing. Was it outsourced?

Lines 399-401 – May cite the reference.

Line 410 – Method of transformation can be mentioned.

Figure 1 – Mark the lanes where the mutants were diagnosed; with some symbol such as *. Similarly , mark important lanes in the gels of all other figures too.

Figure 4 – Seems to be stretched. Figures 4 and 5 can be combined as A, B, C, and D.

The number of figures are too many. The sizes of some can be reduced and combined, and some can go into supplementary.

Minor corrections

Italicize enzyme names

Small grammatical corrections are many across the entire manuscript. For example -

Line 111 - check ‘the GUS is’

Line 362 - check ‘harvest’

Line 119 - check ‘for’

Line 352 - check

Author Response

This manuscript reports a novel and interesting application of treating plants with nicotinamide to enhance genome editing efficiency. Poor editing efficiency is indeed a challenge in many crops, especially polyploids such as wheat. The authors have standardized various concentrations and treatment intervals of nicotinamide. They study the editing efficiency of nicotinamide in wheat callus, as well as mature, and immature embryos. They were able to edit GUS gene with single copy and the TaWaxy that has three copies on three different chromosome. It is appreciable indeed that the authors achieved enhanced editing efficiencies. However, there are some minor suggestions/queries -

Lines 77–79 - Please cite appropriate reference.

Response: Thanks for your suggestion. This sentence and the following are from the same research, and we have cited the reference 20 again after the sentence in line 77-79.

Lines 113-114 – The authors write about 48 and 64 plants, but the image has lesser lanes. Some lanes do not have bands, pl write the explanation in legend as ‘putatively transgenic’ and not ‘transgenic’.

Response: Because the PCR is very easy thing, and there are too many putatively transgenic, so we just show some parts of the results. Some of T0 generation is heterozygous transgenic plant, so the negative plants could be appeared in T1 generation. The “transgenic” had been changed to “putatively transgenic” in the legend of Figure S1a and Figure S3a.

Lines 114-116 – The result is not explained clearly. What is it that the authors want to convey? Please explain.

Response: Because if the transgenic plants contain gene editing vectors, the next generation may produce new mutations. The purpose of mutation detection of the target gene again is to prove that the transgenic plant has no mutation before nicotinamide treatment, so as to ensure that the new mutants were produced by nicotinamide treatment.

Lines 188-194 – The information is unclear. Is it that the authors want to convey that they repeat nicotinamide treatment on lines mutated at one locus in order to achieve mutations at two additional loci. If that is the case, nicotinamide works randomly and I wonder if it should be able to aid editing loci on multiple chromosomes during the first round of 14-day treatment itself.

Response: Yes, nicotinamide works randomly, the results of Table 6 was shown that the effect of nicotinamide treatment in wheat genetic transformation for base editing, the mutation also can occur in two chromosomes.

Lines 311-312 – Cite the reference.

Response: We are very sorry for this mistake. We have cited the reference in Lines 311-312 in the manuscript.

Lines 389-391 – Was the digestion performed after completing the PCR reaction or before?

Response: The PCR reaction was performed before the digestion. So we have revised this sentence into “the specific PCR products were digested at 37 °C for 2 h in the 20 μl reaction buffer consisted of the corresponding restriction enzymes (1 U each) and 10 μl PCR products”.

Line 393 – Check 2%. I wonder if the authors used this concentration to separate bands of sizes 300-500 bp.

Response: We are very sorry for this mistake. The 1% agarose gel was used; we have changed it in the manuscript.

Lines 396-397 – Details of sequencing. Was it outsourced?

Response: Yes, the sequencing was performed in Sangon Biotech (Shanghai, China). We had provided this information in Lines 396-397.

Lines 399-401 – May cite the reference.

Response: Thanks for your suggestion. We have cited the reference in Lines 399-401.

Line 410 – Method of transformation can be mentioned.

Response: Thanks for your nice suggestion. This sentence had been changed to “The base editing vector was transformed into Agrobacterium strain C58C1 for wheat transformation”.

Figure 1 – Mark the lanes where the mutants were diagnosed; with some symbol such as *. Similarly , mark important lanes in the gels of all other figures too.

Response: Thanks for your nice suggestion. The lane of mutants had been marked with * in the gels figures.

Figure 4 – Seems to be stretched. Figures 4 and 5 can be combined as A, B, C, and D.

The number of figures are too many. The sizes of some can be reduced and combined, and some can go into supplementary.

Response: Thanks for your suggestion. According the suggestion of reviewer 4, Figures 1, 3, 4 and 6 was moved to the supplementary information.

Minor corrections

Italicize enzyme names

Response: Thanks for your nice suggestion. All of the enzyme names had been changed to italics format.

Small grammatical corrections are many across the entire manuscript. For example -

Line 111 - check ‘the GUS is’

Response: Thanks for your nice suggestion. “The GUS is” had been changed to “The GUS gene is” in Line 111.

Line 362 - check ‘harvest’

Response: We are sorry for this mistakes. The “harvest” had been changed to “harvested”.

Line 119 - check ‘for’

Response: Thanks for your nice suggestion. We have changed “treated for” to “treated in” in line 119.

Line 352 – check

Response: Thanks for your nice suggestion. We had revised this sentence.

Reviewer 4 Report (New Reviewer)

The core of the manuscript is the application of a simple yet effective approach to increase the editing frequency of multiple gene loci in polyploid common wheat using nicotinamide as a tissue culture component. The approach described by the authors can help increase the efficiency of gene editing and the number of independent events, which is still a difficult task for plant biotechnologists working with wheat. It is an interesting and promising methodological manuscript.  

My comments are as follows:

 The Title

Authors should indicate, what kinds of wheat was the subject of the experiments, ‘hexapliod wheat’ or ‘T.aestivum L.’ should be added.

Abstract

Please correct phrases in lines 17 -21, unclear. There is a need for a careful formulation.

Results:

Lines 110-114. This part needs correction. In experiment 2.1 wheat embryos were taken from double transgenic parent plants, that carry an active GUS gene (homozygous, first transformation) and inactive genome editing sequences (hemizygous, second transformation). Modifications should be also made for phrases in abstracts (lines 17-21), explanation of the tables 1 (see comments below) and description of plant materials (lines 357-360). 

Line 147, replace ‘mutants’ with ‘mutant’, as the one biallelic plant is described in the text.

Line 159. Correct ‘niacinamide’

Lines 188-190. Need correction, unclear meaning

Line 279 correct 2.5 mM  

Line 282 correct sgRNA (not SgRNA)

DISCUSSION

Lines 319-321 The statement ‘5) Different plant species. Different plant species may also lead to different effects of nicotinamide treatment’ It’s reasonable, but I would like to see more info, like it was done in others statements. Phrase ‘Different plant species’ is repeated twice, confusing reading. 

Line 322 correct sgRNA (not SgRNA)

Materials and methods

Section 4.1

Reference 38 does not describe transgenic wheat plants transformed with pMWB110-SpCas9-TaU3-GUS (genotype: H29) or pMWB110-SpCas9-TaU3-Waxy (lines 357-360). Please provide an appropriate reference to the origin of transgenic wheat plants.

Section 4.2.

Indicate the provider of the nicotinamide (line 373)

Section 4.4

Lines 406 and 408, correct sgRNA (not SgRNA)

Section 4.5

there is no indication when nicotinamide was added to the culture medium during transformation.

Figures

I suggest to move Figures 1, 3, 4 and to the additional materials. The gels are only examples describing the technical details of the detection of mutant plants. Since the edited sequences and spectrograms are also included as figures, in my opinion this is more informative and quite sufficient for the main part.

Figure 2

line 151. I recommend replacing "mutants" with "mutant wheat plants" in the title.

Line 152. Am I correct in understanding that "H29" is "non-transgenic variety H29"? Refine it.

Line 154. Please, correct "IV4: GUS gene transgenic plants in H29", I also suggest to replace "biallelic mutants" with "plant with biallelic mutations".

Line 155. 'InDel’ means insertion and deletion, on Figure 2b there is no new mutant plant with insertions, only deletions. Please, correct.

 Tables

Table 1

I think the title (or footnotes) should state that the embryos were taken from wheat plants harboring components for editing GUS gene, while the GUS gene was introduced separately prior to transformation with the gene editing sequences.

Please correct efficiency in the table from 17.0 to 16.7.

Tables 2, 3 and 4

I recommend changing the title (or footnotes) to indicate that the embryos were derived from wheat plants with gene editing components.

What does "Detected Plants" mean? “positive plants” are transgenic seedlings that have inherited gene editing sequences? Please include missing information to better understand the results.

 Table 5

I think the title could be modified, like ‘Effect of nicotinamide on the efficiency of Agrobacterium-mediated genetic transformation of wheat’. The footnote could indicate the name of the wheat variety and the stage of tissue culture at which nicotinamide was added to the culture medium.

Table 6

 Do the authors have an explanation for the appearance of a new mutation in one of the offspring at the TaWaxy-A loci in the absence of nicotinamide treatment?

 General comment

In my opinion, there are not enough repetitions; there is no statistical data processing. Explanations are needed.

Author Response

The core of the manuscript is the application of a simple yet effective approach to increase the editing frequency of multiple gene loci in polyploid common wheat using nicotinamide as a tissue culture component. The approach described by the authors can help increase the efficiency of gene editing and the number of independent events, which is still a difficult task for plant biotechnologists working with wheat. It is an interesting and promising methodological manuscript. 

My comments are as follows:

 The Title

Authors should indicate, what kinds of wheat was the subject of the experiments, ‘hexapliod wheat’ or ‘T.aestivum L.’ should be added.

Response: Thanks for your nice suggestion. The title had been changed to “Application of Nicotinamide in Culture Medium Improves the Efficiency of Genome Editing in hexapliod Wheat”.

Abstract

Please correct phrases in lines 17 -21, unclear. There is a need for a careful formulation.

Response: Thanks for your nice suggestion.

Results:

Lines 110-114. This part needs correction. In experiment 2.1 wheat embryos were taken from double transgenic parent plants, that carry an active GUS gene (homozygous, first transformation) and inactive genome editing sequences (hemizygous, second transformation). Modifications should be also made for phrases in abstracts (lines 17-21), explanation of the tables 1 (see comments below) and description of plant materials (lines 357-360).

Response: Thanks for your nice suggestion. We had provided this information in lines 17-21, table 1 and lines 357-360.

Line 147, replace ‘mutants’ with ‘mutant’, as the one biallelic plant is described in the text.

Response: Thanks for your nice suggestion. We had changed “mutants” to “mutant”.

Line 159. Correct ‘niacinamide’

Response: We are sorry for this mistakes. We had changed “niacinamide” to “nicotinamide”.

Lines 188-190. Need correction, unclear meaning

Response: Thanks for your suggestion. We have deleted this sentence.

Line 279 correct 2.5 mM 

Response: Thanks for your nice suggestion. We had changed “2.5mM” to “2.5 mM”.

Line 282 correct sgRNA (not SgRNA)

Response: Thanks for your nice suggestion. We had changed “SgRNA” to “sgRNA”.

DISCUSSION

Lines 319-321 The statement ‘5) Different plant species. Different plant species may also lead to different effects of nicotinamide treatment’ It’s reasonable, but I would like to see more info, like it was done in others statements. Phrase ‘Different plant species’ is repeated twice, confusing reading.

Response: Thanks for your nice suggestion. We had changed “5) Different plant species” to “5) Species”.

Line 322 correct sgRNA (not SgRNA)

Response: Thanks for your nice suggestion. We had changed “SgRNA” to “sgRNA”.

Materials and methods

Section 4.1

Reference 38 does not describe transgenic wheat plants transformed with pMWB110-SpCas9-TaU3-GUS (genotype: H29) or pMWB110-SpCas9-TaU3-Waxy (lines 357-360). Please provide an appropriate reference to the origin of transgenic wheat plants.

Response: We are very sorry for this mistakes. We have had changed “reference 38” to “reference 8”.

Section 4.2.

Indicate the provider of the nicotinamide (line 373)

Response: Thanks for your nice suggestion. The information of nicotinamide (N0636, Sigma) had been provided in section 4.2.

Section 4.4

Lines 406 and 408, correct sgRNA (not SgRNA)

Response: Thanks for your nice suggestion. We had changed “SgRNA” to “sgRNA”.

Section 4.5

there is no indication when nicotinamide was added to the culture medium during transformation.

Response: Thanks for your nice suggestion. The nicotinamide was added in WLS-Res medium; we had provided this information in section 4.5.

Figures

I suggest to move Figures 1, 3, 4 and to the additional materials. The gels are only examples describing the technical details of the detection of mutant plants. Since the edited sequences and spectrograms are also included as figures, in my opinion this is more informative and quite sufficient for the main part.

Response: Thanks for your nice suggestion. We had moved Figures 1, 3, 4 and 6 to the additional materials.

Figure 2

line 151. I recommend replacing "mutants" with "mutant wheat plants" in the title.

Response: Thanks for your nice suggestion. We had changed “mutants” to “mutant wheat plants”.

Line 152. Am I correct in understanding that "H29" is "non-transgenic variety H29"? Refine it.

Response: Thanks for your nice suggestion. We had changed “H29” to “non-transgenic variety CB037”.

Line 154. Please, correct "IV4: GUS gene transgenic plants in H29", I also suggest to replace "biallelic mutants" with "plant with biallelic mutations".

Response: Thanks for your nice suggestion. We had changed “GUS gene transgenic plants in H29” to “the transgenic line 29 with GUS gene”, and “biallelic mutants” to “plant with biallelic mutations”.

Line 155. 'InDel’ means insertion and deletion, on Figure 2b there is no new mutant plant with insertions, only deletions. Please, correct.

Response: Thanks for your nice suggestion. We had changed “InDel” to “Deletion”.

 Tables

Table 1

I think the title (or footnotes) should state that the embryos were taken from wheat plants harboring components for editing GUS gene, while the GUS gene was introduced separately prior to transformation with the gene editing sequences.

Response: Thanks for your nice suggestion. We had provided the information “the double transgenic parent plants which carry an active GUS gene (homozygous, first trans-formation) and genome editing vectors (hemizygous, second transformation)” in the title of table 1.

Please correct efficiency in the table from 17.0 to 16.7.

Response: We are very sorry for this mistakes. We have had changed “17.0” to “16.7”.

Tables 2, 3 and 4

I recommend changing the title (or footnotes) to indicate that the embryos were derived from wheat plants with gene editing components.

Response: Thanks for your nice suggestion. We had provided the information in the titles of tables 2, 3, and 4.

What does "Detected Plants" mean? “positive plants” are transgenic seedlings that have inherited gene editing sequences? Please include missing information to better understand the results.

Response: Thanks for your nice suggestion. We had provided the missing information in the tables 2, 3, and 4.

 Table 5

I think the title could be modified, like ‘Effect of nicotinamide on the efficiency of Agrobacterium-mediated genetic transformation of wheat’. The footnote could indicate the name of the wheat variety and the stage of tissue culture at which nicotinamide was added to the culture medium.

Response: Thanks for your nice suggestion. The title of table 5 had been changed to “Effect of nicotinamide on the efficiency of Agrobacterium-mediated genetic transformation of wheat “; and the “Note: The genotype was Fielder and the nicotinamide was added in WLS-Res medium” also provided in the footnote of table 5.

Table 6

 Do the authors have an explanation for the appearance of a new mutation in one of the offspring at the TaWaxy-A loci in the absence of nicotinamide treatment?

Response: There are some researches about that if the transgenic plants carry the genome vector, the new mutation can be generated in the next generation in other species (for example Arabidopsis, rice and maize). But many genes had been knockout by CRISPR in our lab, we found that it is very difficult to generate new mutation in the next generation in wheat. We think that the reason may be the huge genome or the different insert sites of Cas9.

 General comment

In my opinion, there are not enough repetitions; there is no statistical data processing. Explanations are needed.

Response: We use the transgenic seeds, so the amount of seeds is relatively small. It is very difficult to do repetitions. And in order to confirm the effect of nicotinamide, we used different tissues to verify. Please you can understand.

This manuscript is a resubmission of an earlier submission. The following is a list of the peer review reports and author responses from that submission.

Round 1

Reviewer 1 Report

The manuscript entitled “Application of nicotinamide in culture medium improves the efficiency of genome editing in wheat” stated that the addition of nicotinamide in media is able to increase genome editing efficiency in wheat. Authors firstly checked nicotinamide function in editing efficiency for gus gene; and the wheat waxy gene was used to investigate this chemical role in gene editing efficiency in immature and mature embryos. This research is able to promote wheat genome editing. Nevertheless, the manuscript quality required to be improved to fit this journal.

Major concerns:

1.     The report (Tiricz, H.; Nagy, B.; Ferenc, G.; Torok, K.; Nagy, I.; Dudits, D.; Ayaydin, F. Relaxed chromatin induced by histone deacetylase inhibitors improves the oligonucleotide-directed gene editing in plant cells. J. Plant Res. 2018, 131, 179-189.) weakens the advance of this manuscript.

2.     There are no mechanisms underlying nicotinamide roles in genome editing presented in this study.

3.     The gus gene editing mutants required to be shown by GUS staining.

4.     The editing plants required be sequenced to perform genotyping to strengthen the results.

Author Response

  1. The report (Tiricz, H.; Nagy, B.; Ferenc, G.; Torok, K.; Nagy, I.; Dudits, D.; Ayaydin, F. Relaxed chromatin induced by histone deacetylase inhibitors improves the oligonucleotide-directed gene editing in plant cells. J. Plant Res. 2018, 131, 179-189.) weakens the advance of this manuscript.

Response: Thanks for your advice. It is a new attempt that nicotinamide was used to increase the efficiency of CRISPR in plants. There are many similar research reports in human cells, but there is really no more suitable report in plants than this one because there are no other research reports. Please you can understand.

  1. There are no mechanisms underlying nicotinamide roles in genome editing presented in this study.

Response: Thanks for your suggestion. Wheat seedlings were treated with nicotinamide, and then transcriptome analyzed found that nicotinamide treatment would change the expression of chromatin state related gene, such as the down-regulation expression of six methyltransferase synthesis pathway genes. But this section has been published in Dai et al. 2021, and we have supplemented this reference in induction in lines 227-229.

  1. The gus gene editing mutants required to be shown by GUS staining.

Response: Thanks for your nice suggestion. We have supplemented the GUS staining results in Figure 4a.

  1. The editing plants required be sequenced to perform genotyping to strengthen the results.

Response: Thanks for your good suggestion. We have supplemented the sequencing results of GUS and TaWaxy gene mutants in Figure 4b and 4c.

Reviewer 2 Report

In this study, conducted by Wang and colleagues, authors analyze the effect if nicotinamide, a histone deacetylase inhibitor, on the edition efficiency of Cas9. They determine that nicotinamide treatment increases the number of mutants, especially at 7 and 14 days.

Even these results open the possibility to improve genome edition efficiency using this strategy, there several major points:

- figure legends must be improved.

- table 1 is not in the correct place.

- legend to figure 3 does not match (legend indicates panels a, b, c , and d but there are only 3 panels).

- materials and methods must be improved: authors indicate they detect positive T1 plants by bar gene amplification, but they do not indicate why this amplification detects the positive ones. Which restriction enzyme sites they mutate for each gene and which are the sizes of the expected fragments with or without the mutation? They do not explain why they analyze both GUS and  TaWaxy genes.

- english language and style must be improved (check the abstract for instance).

Therefore, I would not recommend its publication unless its send back for reconsideration after major revisions are done. 

Author Response

In this study, conducted by Wang and colleagues, authors analyze the effect if nicotinamide, a histone deacetylase inhibitor, on the edition efficiency of Cas9. They determine that nicotinamide treatment increases the number of mutants, especially at 7 and 14 days.

Even these results open the possibility to improve genome edition efficiency using this strategy, there several major points:

  • figure legends must be improved.

Response: Thanks for your suggestion. We have improved every figure legend in this manuscript.

2- table 1 is not in the correct place.

Response: We are very sorry for this mistake; we have moved the table 1 to lines 527-530.

3- legend to figure 3 does not match (legend indicates panels a, b, c , and d but there are only 3 panels).

Response: We are very sorry for this mistake; the PCR detection of bar gene is very simple, so we deleted this picture and we forgot to deleted the legends. Now we have deleted the legends about PCR detection of bar gene in figure 3.

4- materials and methods must be improved: authors indicate they detect positive T1 plants by bar gene amplification, but they do not indicate why this amplification detects the positive ones.

Response: Thanks for your nice suggestion. The premise that nicotinamide treatment can work is that there must be CRISPR vector in the plant, and generally transgenic plants are heterozygous, which will appear negative plants in the next generation. Therefore, we must ensure that the treated plants contain gene editing vectors by bar gene detection. We have explained the reason of bar gene detection in 4.3 section of materials and methods in lines 425-426.

5-Which restriction enzyme sites they mutate for each gene and which are the sizes of the expected fragments with or without the mutation?

Response: Thanks for your nice suggestion. The restriction enzymes of SnaBI and BglII were used to detect the GUS and TaWaxy gene mutants, respectively. This information had been supplemented in in 4.3 section of materials and methods and figure legends.

Three types of band patterns were found in the PCR-RE assay: heterozygous monoallelic mutants gave three bands, biallelic mutants gave only the biggest band, and non-mutants as well as wild-type (WT) plants gave two completely digested bands. This information also had been supplemented in in 4.3 section of materials and methods.

6-They do not explain why they analyze both GUS and TaWaxy genes.

Response: Thanks for your nice suggestion. After all, GUS gene is an exogenous gene. We want to further repeat and confirm the effect of nicotinamide on endogenous genes. We have explained this reason in 2.2 section in lines 533-534.

7- english language and style must be improved (check the abstract for instance).

Response: Thanks for your nice suggestion. The manuscript had been improved by commercial company, and provided the confirmation certificate.

Reviewer 3 Report

Genome editing efficiency in wheat has always been a problematic topic for plant biotechnologists. With this regards, this research article shows improvement in CRISPR based genome editing in wheat. 

I do have certain serious concerns about the presentation of data. Most of the presented data on mutations is based on gel images rather than actual sequencing results. Therefore, I do not trust the data just based on gel images. Authors should include sequencing data including quality reads from sequencing. Even papers in reputed journals shows some sequencing data. 

In abstract, authors mention in Line 16-17 that they do editing efficiency is affected by chromatin status. This statement lacks scientific maturity and more elaborate information must be provided on chromatin and histones. 

Authors state in Line 20, that they see 36% editing efficiency. Later they claim similar efficiency was observed in immature embryos which is way less than 36%. Is it a typing error or authors correlate it to something else?

What is the role of TaWaxy gene? Authors suddenly jump to a random gene without any background information in abstract. 

Further, many basic scientific errors -

Irregular font sizes

Irregular reference formatting could be observed. 

Based on the data presented, I do not find sufficient evidence on editing efficiency due to lack of sequencing data. Unless information on sequencing is provided in detail, I would recommend to reject the article. 

Author Response

1、I do have certain serious concerns about the presentation of data. Most of the presented data on mutations is based on gel images rather than actual sequencing results. Therefore, I do not trust the data just based on gel images. Authors should include sequencing data including quality reads from sequencing. Even papers in reputed journals shows some sequencing data. 

Response: Thanks for your nice suggestion. The sequencing results of GUS, TaWaxy and base editing were supplemented in Figure 4b, 4c and 7, respectively. Moreover, we have supplemented the GUS staining results in Figure 4a to confirm the GUS gene loss the function in GUS mutants.

2、In abstract, authors mention in Line 16-17 that they do editing efficiency is affected by chromatin status. This statement lacks scientific maturity and more elaborate information must be provided on chromatin and histones. 

Response: Thanks for your nice suggestion. This result was concluded by previously study, we have deleted the statement to avoid misunderstanding.

3、Authors state in Line 20, that they see 36% editing efficiency. Later they claim similar efficiency was observed in immature embryos which is way less than 36%. Is it a typing error or authors correlate it to something else?

Response: We are very sorry that the sentences were not very clear. The 36% editing efficiency was achieved for GUS gene, and a similar efficiency 30.3% was reached for TaWaxy gene. So the name of GUS gene was supplemented in line 19, and the similar efficiency was changed into “The editing efficiency could increase to 30.3%” in lines 23-24.

4、What is the role of TaWaxy gene? Authors suddenly jump to a random gene without any background information in abstract. 

Response: Thanks for your nice suggestion. The endogenous TaWaxy gene determined the synthesis of amylose, and we have supplemented the role of TaWaxy gene in line 21-22.

5、Further, many basic scientific errors -

Irregular font sizes

Response: We are very sorry for this mistake; we have changed the font sizes.

6、Irregular reference formatting could be observed. 

Response: We are very sorry for this mistake, we have carefully revised the reference formatting again.

7、Based on the data presented, I do not find sufficient evidence on editing efficiency due to lack of sequencing data. Unless information on sequencing is provided in detail. 

Response: The sequencing results of GUS, TaWaxy and base editing were supplemented in Figure 4b, 4c and 7, respectively. Moreover, we have supplemented the GUS staining results in Figure 4a to confirm the GUS gene loss the function in GUS mutants.

Round 2

Reviewer 1 Report

The revised manuscript entitled “Application of nicotinamide in culture medium improves the efficiency of genome editing in wheat” had a good improvement in quality. Authors added some experiments including GUS staining and DNA sequencing as well as response to comments. No comments should be provided now.

Reviewer 2 Report

The present version includes all the referees' comments. I recommend this work for publication.